# Course of Metal Ions after a Revision of Malfunctioning Metal-on-Metal Total Hip Prostheses

**DOI:** 10.3390/medicina57020115

**Published:** 2021-01-28

**Authors:** Annamaria Nicolli, Isabella Bortoletti, Stefano Maso, Andrea Trevisan

**Affiliations:** Department of Cardiac Thoracic Vascular Sciences and Public Health, University of Padova, Via Giustiniani 2, 35128 Padova, Italy; annamaria.nicolli@unipd.it (A.N.); isabella.bortoletti@aopd.veneto.it (I.B.); stefano.maso@unipd.it (S.M.)

**Keywords:** metal-on-metal hip prostheses, metal ions, hip revision, cobalt, chromium

## Abstract

The present research evaluated the course of cobalt and chromium in the blood and urine after the revision of metal-on-metal with a ceramic-on-polyethylene total hip arthroplasty. Seven patients were enrolled for hip prosthesis revision owing to ascertained damage of the implant. Metals in the blood and urine were evaluated before and after the hip revision. The double measurement before the total hip revision revealed high levels of metal ions (on average, 88.1 µg/L of cobalt in the blood, 399.0 µg/g of creatinine cobalt in the urine, 46.8 µg/L of chromium in the blood, and 129.6 µg/g of creatinine chromium in the urine at the first measurements), with an increasing trend between the first and second dosage. Within a week after the hip revision, the levels of metal ions significantly decreased by approximately half. Four to six months after the operation, the cobalt levels were found near to the reference values, whereas the chromium levels reached 25% of the values measured before the revision. The revision of malfunctioning metal-on-metal implants produced a dramatic decrease of metal ions in biological fluids, although it did not completely rescue the chromium level.

## 1. Introduction

A consequence of metal ion debris accumulation in periprosthetic tissues caused by malfunctioning metal-on-metal (MoM) total hip prostheses (THPs) is so-called metallosis [1]. Cobalt (Co) and chromium (Cr) accumulation is an undesirable feature that can result from implant wear [2,3,4]. On the other hand, metal-on-polyethylene bearings also cause a significant increase in metal ions [5]. The increase in metal ions is generated by corrosive action in the synovial fluid of accumulated metal debris [6], inducing macrophage activation, osteolysis, and ultimately, implant malfunctioning. The Medicines and Healthcare Products Regulatory Agency (MHRA) in Great Britain states that Co and Cr concentrations in whole blood below 7 µg/L are acceptable [7].

The toxic effects as a consequence of metal ion release are almost exclusively caused by Co (so-called arthroprosthetic cobaltism). The toxicity of Co is due to a number of involved organs and tissues, such as the thyroid (hypothyroidism due to the inhibition of tyrosine iodinase in the gland) [8], heart (congestive heart failure) [9], and cranial nerves (severe auditory and optic nerve toxicity) [10]. On the other hand, systemic adverse effects typically occur with blood Co concentrations higher than 300 µg/L, with polycythemia and hypothyroidism being the first signs of toxicity [11].

The only approach used to prevent the toxic effects of Co is the revision of a malfunctioning prosthesis because, while the use of chelating therapy with ethylenediaminetetraacetic acid disodium calcium [12] or 2,3-dimercaptopropane-1-sulfonate [13] could be of some effectiveness, it is not advisable due to its severe side effects [14].

The aim of the present research was to study the course of metal ions after the revision of malfunctioning MoM THPs with ceramic-on-polyethylene (CoP) THPs.

## 2. Materials and Methods

### 2.1. Patients

From a population of 72 patients with MoM THPs with a large diameter femoral head ASR™ DePuy (Raynham, MA, USA) and 1 patient with hip resurfacing subjected to a device alert recall program, as previously studied [2], 19 (26%) were judged eligible for an MoM prosthesis revision according to the malfunctions of their hips based on the pain score (Harris Hip Score), X-ray evaluation, inclination angle of the cup, and high metal ion levels in biological fluids. Of these 19 subjects, 7 (all with THPs) were included in this study because they had performed at least two controls before replacing the malfunctioning implants and, in particular, had followed the protocol of control of the metal ions that included an assessment at 7 days and 4 to 6 months after the revision. 

They were five males and two females, aged, on average, 57.3 years (range 38–67 years) at the time of the total MoM hip arthroplasty and 62.7 years (range 45–72 years) at the time of the revision of the malfunctioning prosthesis. The interval between the surgeries was, on average, 5.4 years (range 4–7 years). The hip revision was performed with CoP prostheses leaving the same stem in situ and changing the socket, insert, and head. The head was ceramic with a metal ring inserted (sleeve) to prevent ceramic damage. For comparison, the metal ion levels of a group of 27 patients with well-functioning MoM THPs belonging to the same cohort of patients covered by this research are shown.

The patients gave their informed consent to surgery and anonymous treatment of their data.

### 2.2. Analysis of Metals

The whole blood (CoB and CrB, respectively) and urine (CoU and CrU, respectively) level of cobalt and chromium, respectively, was measured two times, namely, at the recall (time 0) and three months after. Once the implant was revised with a CoP prosthesis, the metal ion levels were measured seven days and 4 to 6 months later using a graphite furnace atomic absorption spectrophotometer Perkin-Elmer (Waltham, MA, USA) AAnalyst 600 with a Zeeman effect background correction. The detection limits of the metals were as follows: CoB and CoU 0.56 µg/L, CrB 0.1 µg/L, and CrU 0.08 µg/L. The reference values in not occupationally exposed people and those not subjected to THP were as follows: CoB 0.055–0.44 µg/L, CoU 0.077–2.2 µg/L, CrB 0.06–1.1 µg/L, and CrU 0.05–06 µg/L. The reagents and standard solutions used were high-purity Atomic Absorption Spectrometry (AAS) grade standard stock (Perkin-Elmer, Waltham, MA, USA). The urinary values were adjusted to creatinine according to our previous study [2], which were determined using the basic picrate Jaffe reaction.

### 2.3. Statistics

The nonparametric (Wilcoxon signed ranks) test was applied to compare the whole blood metal ions before and after the hip revision. Statistical significance was considered to be *p* < 0.05. Statsdirect 2.7.7 version (Statsdirect Ltd., Birkenhead, Merseyside, UK) was used for the statistical analyses.

## 3. Results

Six out of the seven patients who submitted to a revision of an MoM THP according to the criteria explained above had an inclination angle of the MoM prosthesis greater than 50° (57 ± 6°); 45° was the inclination angle of the seventh patient. No symptom of Co toxicity was shown by the patients, and no chelating therapy was adopted.

Two measurements of the metal ion levels in whole blood and urine before the MoM total hip revision (at the recall, i.e., time 0, and after three months) showed an increasing trend (on average, 88.1 µg/L for CoB, 399.0 µg/g for creatinine CoU, 46.8 µg/L for CrB, and 129.6 µg/g for creatinine CrU at time 0; 91.2 µg/L for CoB, 515.5 µg/g for creatinine CoU, 48.1 µg/L for CrB, and 213.7 µg/g for creatinine CrU after three months); however, these increases were not statistically significant.

Within a week after the MoM hip revision with the CoP prostheses, the level of metal ions significantly (*p* < 0.05) dropped (on average) to 32.2% for CoB (29.4 µg/L), 32.5% for CoU (167.6 µg/g of creatinine), 56.8% for CrB (27.3 µg/L), and 34.2% for CrU (73.0 µg/g of creatinine).

After four to six months, the CoB and CoU levels were near the reference values, at 2.1% for CoB (1.9 µg/L on average) and 1% for CoU (5.4 µg/g of creatinine). Regarding the Cr ions, the CrB levels were at 22.7% (10.9 µg/L) and 10.4% for CrU (22.3 µg/g of creatinine). The results are summarized in Figure 1 (Co) and Figure 2 (Cr).

In Table 1, the metal ion levels of patients with well-functioning MoM hip prostheses belonging to the same cohort of patients covered by this research are shown for comparison.

## 4. Discussion

A non-negligible number of MoM prostheses (one in eight) are destined for revision within 10 years [15]. Metal debris released by an MoM prosthesis is a serious problem for a total hip arthroplasty [16,17]. High levels of metal ions can cause severe toxic effects [18,19].

In general, an increase in blood metal ions is significant within five years after surgery [20], although it has been proven that, in a well-functioning total hip arthroplasty, serum Co levels can remain low after 21 years [21]. Resurfacing is at a greater risk of increasing the metal ions compared with THP [22,23], although trunnion wear or corrosion could cause a greater increase of Co, but not Cr, in a total hip arthroplasty [24]. Finally, the presence of pseudotumors is related to higher levels of metal ions, and physical activity positively influences the metal ion levels and pseudotumor formation [25].

For this reason, patients with high metal ion values and, mostly, with signs of prosthesis malfunction, should undergo revision. In the present research, patients with a malfunctioning MoM prosthesis revised using a CoP prosthesis were studied to understand the behavior of metal ions after revision and accurate cleaning of periprosthetic tissues. The metal ions in biological fluids showed a dramatic drop within a short period.

To our knowledge, only two studies approached the behavior of metal ions after MoM hip prosthesis revision [12,26]. Both described the metal behavior in only one patient, measuring the Co and Cr concentrations in whole blood or serum one [12] or two years [26] after the prosthesis revision. The concentration of Co and Cr dramatically decreased, even if, in the latter [26], the metal levels persisted higher that the MHRA criteria, whereas, in the former [12], the Co in serum was as low as the reference values, but the Cr in the whole blood persisted above these values (at 33.2% of the values measured before replacement).

Conversely, a third study [27] reported data for 23 patients one year after revision for the metallosis of an MoM THP with metal-on-polyethylene or ceramic-on-ceramic bearings; the Co in serum and Cr in whole blood returned to almost normal levels. Finally, a fourth study [28] observed consistent substantial decreases of Co and Cr after a hip revision (it is not specified whether this was for whole blood or serum) near to the reference values for thirteen patients. Unfortunately, the study provided no information regarding the metal measurements post-revision.

A limitation of our study is the small number of patients (seven); however, they were accurately studied at a standardized time after the revisions. According to the protocol, patients with high levels of metal ions were monitored every three months and, at surgery, they were not always double-checked.

Almost immediately after revision, the metal ion levels dramatically decreased to about 30–50% of the pre-revision values. Four to six months later, the CoB and CoU were reduced to 2.1% and 1.0%, respectively, of the pre-revision values (near to the reference values), whereas the CrB was surprisingly high at 22.7%, with the CoU at 10.4%, far from the reference values.

A possible explanation for this different metal ion behavior may rely on the fact that the decrease in albumin binding capacity could lead to a greater amount of the free form of Co in the blood [29], which is associated with a rapid transport out from the joint [30]. On the other hand, 10% of Co was retained after one year, and a certain amount is permanently retained [31]. On the contrary, the delay in the Cr elimination could be related to its storage in the tissue [31] and red blood cell trapping [31]. In addition, the accumulation of metal particles in the liver and spleen [32] could contribute to maintaining the level of metal ions, mostly Cr, over the reference value for a time longer than the typical kinetics.

## 5. Conclusions

Although studied for only seven patients, the revision of malfunctioning MoM implants produced a dramatic decrease in metal ions in the whole blood within six months, although this did not reach the reference range. In particular, Cr persisted at higher levels than Co, owing to the different kinetics of these metals. To our knowledge, this is the first study that evaluated the behavior of Co and Cr after the removal of the prosthesis and while taking into consideration the urinary excretion of metals.

The aim of replacing malfunctioning prostheses is to prevent the toxic effects of metal ions. However, it is important to point out that high levels of metal ions are not always a condition for revising the prosthesis, which should be considered only in the case of constructive or surgical problems; if the prostheses are functioning well, it is preferable to frequently monitor the metal ions and the functioning of target organs instead of facing a highly demolishing surgery. It is a question of a risk/benefit assessment [33].

However, well-functioning prostheses require careful periodic surveillance (at least annually) both from the toxicology (evaluation of metal ions) and the orthopedic points of view, with close collaboration between these two specialists.

## Figures and Tables

**Figure 1 medicina-57-00115-f001:**
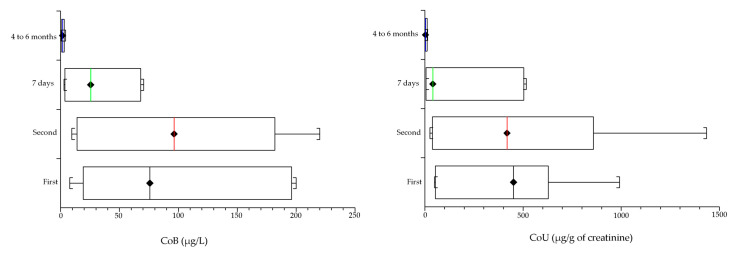
Behavior of Co ions in the whole blood (CoB) and urine (CoU) before and after the revision of the malfunctioning metal-on-metal (MoM) total hip prosthesis (THP): first and second are the measurements carried out before the hip revision. Seven days and 4 to 6 months are the elapsed times since the revision. Within a week after the MoM hip revision with the CoP prostheses, the level of Co significantly (*p* < 0.05) dropped (on average) to 32.2% for CoB (29.4 µg/L) and 32.5% for CoU (167.6 µg/g of creatinine. After four to six months, the CoB and CoU levels were near the reference values, at 2.1% for CoB (1.9 µg/L on average) and 1% for CoU (5.4 µg/g of creatinine).

**Figure 2 medicina-57-00115-f002:**
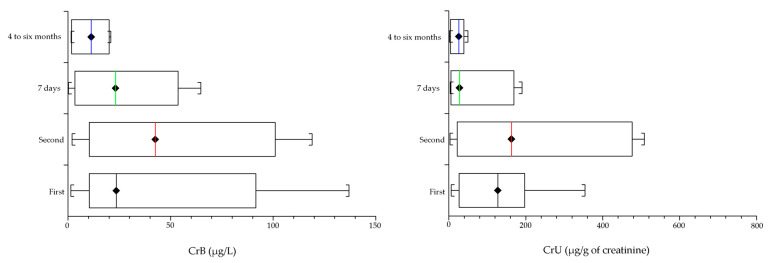
Behavior of Cr ions in whole blood (CrB) and urine (CrU) before and after the revision of the malfunctioning MoM THP: first and second are the measurements carried out before the hip revision. Seven days and 4 to 6 months are the elapsed times since the revision. Within a week after the MoM hip revision with the CoP prostheses, the level of Cr significantly (*p* < 0.05) dropped (on average) 56.8% for CrB (27.3 µg/L), and 34.2% for CrU (73.0 µg/g of creatinine). After four to six months, CrB levels were at 22.7% (10.9 µg/L) and 10.4% for CrU (22.3 µg/g of creatinine).

**Table 1 medicina-57-00115-t001:** Metal ions in the blood and urine in 27 patients (5 females and 22 males) with well-functioning MoM hip prostheses belonging to the same cohort of patients covered by this research compared to the seven patients subjected to hip revision.

Parameters	Well-Functioning THP (*N* = 27)	Malfunctioning THP (*N* = 7 *)
	Mean ± SD	Mean ± SD
CoB (µg/L)	2.8 ± 2.4	91.2 ± 84.4
CoU (µg/g of creatinine)	18.9 ± 21.6	515.5 ± 536.5
CrB (µg/L)	1.4 ± 1.0	48.1 ± 46.3
CrU (µg/g of creatinine)	4.3 ± 4.9	213.7 ± 216.0
Age at surgery (years)	53 ± 7	57 ± 10
Age at analysis (years)	57 ± 7	63 ± 10

Legend: THP = Total Hip Prostheses; SD = standard deviation; (* reported data refers to the second analysis before revision).

## Data Availability

Raw data is available from the author upon request.

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
