# Peer review of "Course of Metal Ions after a Revision of Malfunctioning Metal-on-Metal Total Hip Prostheses"

_medicina, 2021, doi:10.3390/medicina57020115_

Round 1
Reviewer 1 Report
Please us revision or exchange instead of "replacement" Within the abstract, „Four - six months“, should be changed to „Four to six months postoperatively" In the abstract: values of how much the preoperative values decreased should be added. The authors wrote: "seven (all with total hip prostheses) were included in this study because they have performed at least two controls before replacing the malfunctioning implants and had followed the protocol of control of the metal ions that included an assessment 7 days and 4-6 months after the revision“ - were in all cases these 4 assessments of metal ions conducted? Which implants of the new cup were used? This should be added. In 2.2: were the metal ions measured directly before the revision? If not, what was the time period of the measurements before the revision Within the discussion information about metal ions after MOM and hip resurfacing from other studies (not only with cases of revision) should be added. Within the discussion the author mentioned the limitation. Within this paragraph the authors wrote "Almost immediately after revision, the metal ion levels dramatically decreased to about 30-50 percent of the pre-revision values. Four-six months later, CoB and CoU were reduced to 2.1% and 1.0%, respectively, of pre-revision values (near to reference values), whereas CrB was surprisingly high, at 22.7%, as well as CoU, at 10.4%, far from the reference values.“ I do not understand why this facts are within the paragraph of the limitation. Another limitation is that the metal ions were not measured direct before the revision. Do the authors have values of a control group of metal ions after a well functioning MOM or hip resurfacing? This should be added. If not this should be added to a limitation.
Author Response
We are grateful to the reviewer for the careful evaluation of the manuscript. We are able to answer his requests point by point.
- Please us revision or exchange instead of "replacement"
Reply: replacement has been replaced by revision in the title and along the text.
- Within the abstract, „Four - six months“, should be changed to „Four to six months postoperatively"
Reply: the text was changed accordingly (also figures).
- In the abstract: values of how much the preoperative values decreased should be added. The authors wrote: "seven (all with total hip prostheses) were included in this study because they have performed at least two controls before replacing the malfunctioning implants and had followed the protocol of control of the metal ions that included an assessment 7 days and 4-6 months after the revision“ - were in all cases these 4 assessments of metal ions conducted? Which implants of the new cup were used? This should be added.
Reply: the required values have been added. According to the protocol, the dosage of metal ions was performed with variable frequency every three or six months. As far as the new plant is concerned, it is ceramic on polyethylene, as in the text.
- In 2.2: were the metal ions measured directly before the revision? If not, what was the time period of the measurements before the revision
Reply: as in the reply to the previous question, the protocol provides the dosage of metal ions every 3 and 6 months. The dosage was not always done immediately before the refill. For this reason this has been added in the limitations (see reply to point 6).
- Within the discussion information about metal ions after MOM and hip resurfacing from other studies (not only with cases of revision) should be added.
Reply: What was requested was added in the discussion (first paragraph).
- Within the discussion the author mentioned the limitation. Within this paragraph the authors wrote "Almost immediately after revision, the metal ion levels dramatically decreased to about 30-50 percent of the pre-revision values. Four-six months later, CoB and CoU were reduced to 2.1% and 1.0%, respectively, of pre-revision values (near to reference values), whereas CrB was surprisingly high, at 22.7%, as well as CoU, at 10.4%, far from the reference values.“ I do not understand why this facts are within the paragraph of the limitation. Another limitation is that the metal ions were not measured direct before the revision.
Reply: Unfortunately, the sentence that begins with: Almost had to break through. For this reason it seems to belong to the limitations and obviously it is not. This typo has been remedied and the limitations in point 4 have been added (discussion, paragraph 4 and 5).
- Do the authors have values of a control group of metal ions after a well-functioning MOM or hip resurfacing? This should be added. If not this should be added to a limitation.
Reply: a table with metal ion values in a group of patients with well-functioning hip prostheses was added at the end of the results and a conclusive phrase is added at the end of conclusions.
Reviewer 2 Report
Title: "FATE OF METAL IONS AFTER REPLACEMENT OF MAL-FUNCTIONING METAL-ON-METAL HIP PROSTHESES"
"Fate" is perhaps the wrong term here and suggest a larger study. "Course" better describes the content of the paper.
Abstract: "Four-six". This number ist misleading: do you mean "forty-six"? "Four-to-six" month?
This is also misleading in the result section.
Author Response
We are grateful to the reviewer for the careful evaluation of the manuscript and the positive comments to our manuscript. We are able to answer his requests point by point.
- "Fate" is perhaps the wrong term here and suggest a larger study. "Course" better describes the content of the paper.
Reply: we are grateful to the reviewer for the suggestion. Fate has been replaced by course in the title and along the text.
- Abstract: "Four-six". This number its misleading: do you mean "forty-six"? "Four-to-six" month?
This is also misleading in the result section.
Reply: we are sorry we have created some confusion. 4 to 6 months was entered instead of 4-6 months along the text and in figures.
Reviewer 3 Report
Thank you to give me the opportunity to review this paper.
I wiould just dress question to the authors with could be of interest tor the orthopedic community:
Is it recommended to proceed to a large syovectomy during revision?
Does this option might have an impact on the fate of ions?
Best regards and happy new year.
Author Response
We are grateful to the reviewer for the careful evaluation of the manuscript and the positive comments to our manuscript. We are able to answer his requests point by point.
I wiould just dress question to the authors with could be of interest for the orthopedic community:
- Is it recommended to proceed to a large synovectomy during revision?
Reply: putting our hands forward since we are toxicologists and not orthopedics, an accurate synovectomy is recommended in fracturing ceramic implants (see Injury 47 (suppl): 116-120, 2016).
- Does this option might have an impact on the fate of ions?
Reply: referring to the previous answer, during tribology caused by MoM metal residues are in the form of nanoparticles, therefore difficult to isolate. Our opinion is that a thorough synovectomy can favorably affect metal ions.
Round 2
Reviewer 1 Report
Dear authors,
thank you for the changes of the manuscript. The discussion should be optimized by discussing more literature.
Please add a little bit more literature about metal ions concentrations is hip arthroplasty / hip resurfacing.
Here a few papers as a suggestion.
33433099
17068698
25132471
26983720
23572350
32435467
31508310
23950923
Best regards
Author Response
Reviewer 1
Comments and Suggestions for Authors
Dear authors,
thank you for the changes of the manuscript. The discussion should be optimized by discussing more literature.
Please add a little bit more literature about metal ions concentrations is hip arthroplasty / hip resurfacing.
Here a few papers as a suggestion.
33433099, 17068698, 25132471, 26983720, 23572350, 32435467, 31508310, 23950923
Reply: We are grateful to the reviewer for the appreciation and new suggestions that we have welcomed and included in the discussion (they are highlighted in yellow).